**Data Availability Statement:** All the data used in this study are publicly available. The summary statistics of cataract and glaucoma in the European

# Causal associations between rheumatoid arthritis, cataract and glaucoma in European and East Asian populations: A bidirectional two-sample mendelian randomization study

**Menghao Teng**[1☯], **Jiachen Wang**[2☯], **Xiaochen Su**[1], **Ye Tian**[3], **Jiqing Wang**[1], **Yingang Zhang**[1]*

**1** Department of Orthopedics, The First Affiliated Hospital of Xi'an Jiaotong University, Xi'an, Shaanxi, China, **2** Department of Joint Surgery, HongHui Hospital, Xi'an Jiaotong University, Xi'an, Shaanxi, China, **3** Healthy Food Evaluation Research Center, West China School of Public Health and West China Fourth Hospital, Sichuan University, Chengdu, Sichuan, China

☯ These authors contributed equally to this work.
* zyingang@mail.xjtu.edu.cn

## Abstract

### Background

Previous studies have indicated a heightened susceptibility to cataract and glaucoma among rheumatoid arthritis (RA) patients, while it remains uncertain whether RA is causally associated with cataract and glaucoma. A two-sample mendelian randomization (MR) analysis was used to investigate the causal associations between RA, cataract and glaucoma in European and East Asian populations.

### Methods

In the European population, genome-wide association study (GWAS) summary statistics for cataract (372,386 individuals) and glaucoma (377,277 individuals) were obtained from the FinnGen consortium (R9), while RA summary data were derived from a meta-analysis of GWAS encompassing 97173 samples. In the East Asian population, summary data for cataract (212453 individuals), glaucoma (212453 individuals), and RA (22515 individuals) were sourced from the IEU Open GWAS project. Inverse-variance weighted (IVW, random-effects) method served as the primary analysis, complemented by MR–Egger regression, weighted median, weighted mode and simple mode methods. Additionally, various sensitivity tests, including Cochran's Q test, MR–Egger intercept, MR pleiotropy Residual Sum and Outlier test and leave-one-out test were performed to detect the heterogeneity, horizontal pleiotropy and stability of the analysis results.

### Results

Following stringent screening, the number of selected instrumental variables ranged from 8 to 56. The IVW results revealed that RA had an increased risk of cataract (OR = 1.041, 95% CI = 1.019–1.064; P = 2.08×10⁻⁴) and glaucoma (OR = 1.029, 95% CI = 1.003–1.057; P =

population were obtained from the FinnGen database at https://r9.finngen.fi/. The summary statistics for rheumatoid arthritis in the European population were obtained from the GWAS catalog at http://www.ebi.ac.uk/. The GWAS summary statistics of East Asian cataract, glaucoma and rheumatoid arthritis were all extracted from the IEU Open GWAS project at http://gwas.mrcieu.ac.uk/.

**Funding:** This work was supported by the Shaanxi Provincial Administration of Traditional Chinese Medicine (No. 2021-04-ZZ-003). Financial support had no impact on the outcomes of this study.

**Competing interests:** The authors have declared that no competing interests exist.

$2.94 \times 10^{-2}$) in European populations, and RA displayed a positive association with cataract (OR = 1.021, 95% CI = 1.004–1.039; P = $1.64 \times 10^{-2}$) in East Asian populations. Other methods also supported those results by IVW, and sensitivity tests showed that our analysis results were credible and stable.

## Conclusions

This study revealed a positive causality between RA and the increased risk of cataract and glaucoma, which provides guidance for the early prevention of cataracts and glaucoma in patients with RA and furnishes evidence for the impact of RA-induced inflammation on ophthalmic diseases.

## Introduction

Cataract and glaucoma constitute the primary causes of blindness and irreversible vision impairment in developing countries and low-income regions [1, 2]. Cataract is a condition characterized by the opacity of the crystalline lens, which obstructs the passage of light through the lens to the retina [3]. Although the exact pathogenesis of cataract formation remains elusive, it is closely associated with oxidative stress and the natural aging process [1, 4, 5]. Glaucoma is a neurodegenerative disease characterized by the loss of retinal ganglion cells (RGCs) [6]. Several factors contribute to RGCs loss in glaucoma, including aging and elevated intraocular pressure [2, 7]. With the aging of the population, the prevalence of cataract and glaucoma has significantly risen. Previous studies indicated that cataract caused visual impairment in 95 million people by 2014 [1], contributing to over 50% of global cases of blindness [8]. In 2020, it was estimated that glaucoma affected 7.96 million individuals and caused blindness in 3.6 million people aged 50 and above [9], with a projected global impact on 111.84 million individuals by 2040 [10]. Therefore, investigating the pathogenesis and early prevention of cataracts and glaucoma remains a central focus of current research.

Rheumatoid arthritis (RA) is a chronic, autoimmune and inflammatory disease characterized by synovial inflammation, hyperplasia, cartilage and bone destruction, which affects multiple tissues and organs, ultimately leading to joint tissue destruction and chronic disability [11]. As of 2016, the global annual incidence rate of RA is 0.03%, with a prevalence rate of 1% [12]. As a chronic systemic autoimmune disease, inflammation, immunity and heredity are considered to play crucial roles in the development of RA [11, 13]. Numerous studies have explored the potential causal associations between RA and the development of cataracts and glaucoma. For instance, Wenyi Jin et al. reported that cataracts could increase the incidence rate of RA by 71.2% in the European population [14], while Kim SH et al. found that RA was associated with a higher risk of open-angle glaucoma in individuals of Korean ancestry [15]. However, these studies are retrospective, which means that reverse causality and various confounding factors may influence the confidence of these results. Therefore, establishing causal associations between RA and cataracts and glaucoma requires additional evidence from genetic studies.

Mendelian randomization (MR) is an epidemiological research method used to investigate the mechanisms of disease occurrence [16]. In MR analysis, exposure is regarded as an intermediate phenotype, and single-nucleotide polymorphisms (SNPs) are selected as instrumental variables (IVs) to examine the causal associations between the exposure phenotype and disease outcome [17]. Since genetic variants are randomly allocated to each individual at conception

and remain unaffected by environmental factors, MR can mitigate the influence of confounding factors [18, 19]. In addition, MR helps minimize the bias stemming from reverse causality, as an individual's genotype is determined at conception and is not affected by disease progression [18, 20]. Widely adopted, MR has been instrumental in exploring causal associations between RA and diverse diseases, spanning cancer [21], inflammatory bowel disease [22], and osteoporosis [23, 24]. This study harnesses genome-wide association study (GWAS) summary data from the FinnGen project, GWAS catalog and IEU Open GWAS project to evaluate the causal associations between RA, cataracts, and glaucoma through a bidirectional two-sample MR analysis.

## Materials and methods

### Study design

This bidirectional MR analysis was based on GWAS summary data, which were sourced from the FinnGen project, the GWAS catalog (http://www.ebi.ac.uk/) and the IEU Open GWAS database (http://gwas.mrcieu.ac.uk/). We selected SNPs through a rigorous selection process to serve as IVs for subsequent MR analysis. Three fundamental assumptions must be met in MR analysis (Fig 1). First, the SNPs chosen as IVs should exhibit a robust association with the exposures. Second, these SNPs should be independent of confounding factors related to the exposures and outcome. Third, the selected SNPs should influence the outcome solely through the exposures, without involving other biological pathways. This MR analysis meticulously adhered to the recommendations by the STROBE-MR [25].

### Data sources

The GWAS summary data were used to perform the bidirectional two-sample MR analysis. The GWAS summary statistics for cataract and glaucoma, based on the European population, were acquired from the FinnGen consortium (R9), which is one of the largest medicine

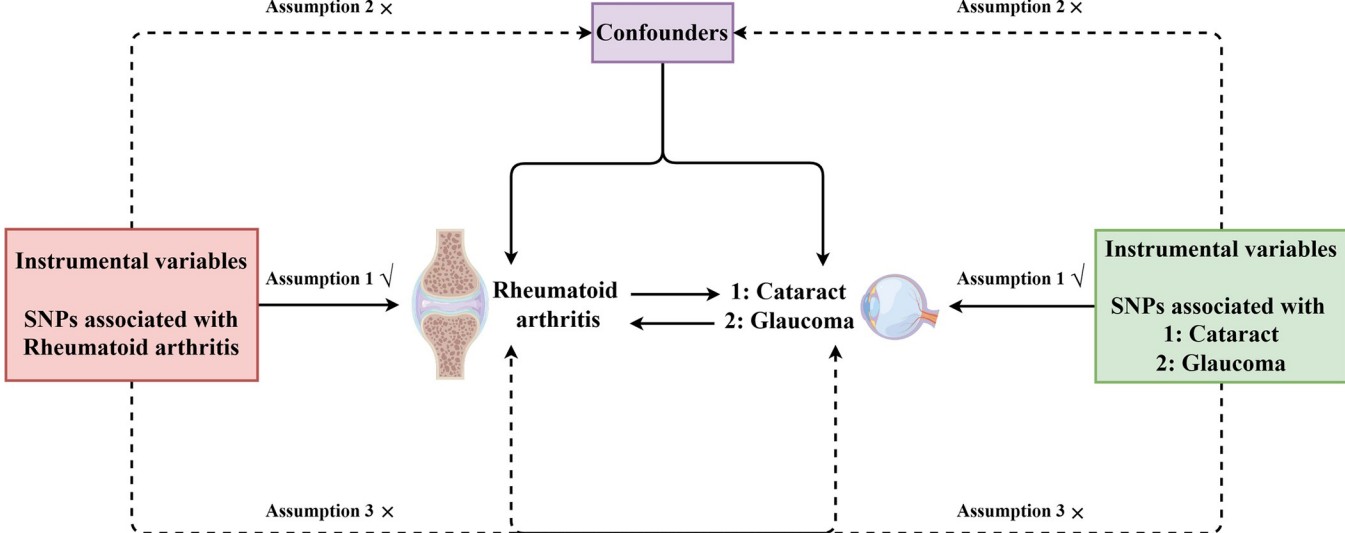

**Fig 1. The diagram of the bidirectional MR study of the causal associations of RA with cataract and glaucoma.** This diagram illustrates the key assumptions that must be satisfied in the bidirectional MR study. First, the SNPs chosen as IVs should exhibit a robust association with the exposures. Second, these SNPs should be independent of confounding factors related to the exposures and outcome. Third, the selected SNPs should influence the outcome solely through the exposures, without involving other biological pathways. IVs: instrumental variables; MR: mendelian randomization; RA: rheumatoid arthritis; SNPs: single-nucleotide polymorphisms.

**Table 1. Detailed information on the summary data for European and East Asian cataract, glaucoma and RA.**

| Population | Trait | Case | Controls | Sample size | SNP (n) | Year | Dataset ID |
|---|---|---|---|---|---|---|---|
| European ancestry | Cataract | 59522 | 312864 | 372386 | 20170144 | 2023 | finngen_R9_H7_ CATARACTSENILE |
| | Glaucoma | 18902 | 358375 | 377277 | 20170237 | 2023 | finngen_R9_H7_ GLAUCOMA |
| | RA | 22350 | 74823 | 97173 | 13297690 | 2022 | ebi-a-GCST-90132223 |
| East Asian ancestry | Cataract | 24622 | 187831 | 212453 | 8885805 | 2019 | bbj-a-94 |
| | Glaucoma | 5761 | 206692 | 212453 | 8885805 | 2019 | bbj-a-121 |
| | RA | 4873 | 17642 | 22515 | 6619872 | 2014 | ieu-a-831 |

SNP (n): the number of single-nucleotide polymorphisms; RA: rheumatoid arthritis.

projects aimed at exploring the genome to improve human health. The cataract summary data encompassed 59,522 cases and 312,864 controls, and the glaucoma summary data consisted of 18,902 cases and 358,375 controls. The summary statistics for RA were derived from a meta-analysis of GWAS encompassing 22,350 cases and 74,823 controls of European ancestry [26].

For East Asian populations, we accessed GWAS summary statistics for cataract, glaucoma and RA from the IEU Open GWAS project (http://gwas.mrcieu.ac.uk/). The summary data included 212,453 samples (24,622 cases and 187,831 controls) for cataract, 212,453 samples (5,761 cases and 206,692 controls) for glaucoma, and 22,515 samples (4,873 cases and 17,642 controls) for RA. Detailed information on the summary data for European and East Asian cataract, glaucoma and RA used in this study is listed in Table 1.

## Selection of IVs

First, in the European population, genome-wide significant SNPs ($P < 5 \times 10^{-8}$) that were strongly associated with the exposures were selected. However, in the East Asian population, the number of SNPs meeting this threshold was not enough to perform the MR analysis when cataract was regarded as the exposure. Therefore, we adopted a higher threshold value ($P < 5 \times 10^{-6}$) to identify relevant IVs in the East Asian population. To ensure the independence of the selected SNPs, we performed a clumping process with $r^2 < 0.001$ and a window size = 10000 kb to eliminate SNPs with strong linkage disequilibrium. Additionally, to minimize weak instrumental bias and guarantee statistical power, SNPs with F statistics < 10 were removed, and the F statistic of each SNP was calculated by the following equation: $F = R^2 \frac{(N-2)}{(1-R^2)}$, where N is the sample size, $R^2$ is genetic variation, and $R^2$ is calculated by the following equation: $R^2 = 2 \times EAF \times (1 - EAF) \times \beta^2$, in which EAF is the effect allele frequency and $\beta$ is the allele effect value. Subsequently, we utilized the PhenoScanner V2 website (www.phenoscanner.medschl.cam.ac.uk) to eliminate SNPs associated with the outcome and confounders. The PhenoScanner V2 is a comprehensive database that collects the associations between human genotypes and phenotypes, and all well-known risk factors for cataract and glaucoma were deemed potential confounders in this study, including ageing, smoking, alcohol consumption, glucocorticoid and steroid usage [27, 28]. In reverse MR analyses, SNPs associated with confounders, including smoking and obesity, were excluded by the PhenoScanner V2 in this step [29]. Finally, to guarantee that the effect alleles were the same allele, we harmonized the exposure and outcome data to remove SNPs that were palindromic with intermediate allele frequencies.

## Statistical analysis

There were five complementary methods employed in this study to estimate the causal associations between RA and cataract and glaucoma, including inverse-variance weighted (IVW,

random-effects), weighted median, MR–Egger regression, simple mode and weighted mode methods. The IVW method served as the primary analysis method because it assumes that all selected IVs are valid and can provide the most accurate results in the absence of horizontal pleiotropy and heterogeneity [30]. MR–Egger regression can provide a consistent causal estimate even if all the genetic IVs are invalid [31], and we were particularly interested in its ability to evaluate horizontal pleiotropy through the intercept value. The weighted median method could provide a relatively precise result, even when up to 50% of the information is derived from invalid IVs [32]. In addition, weighted mode and simple mode were employed as supplementary methods to evaluate the causal associations. The statistical power of the MR analysis was calculated by the mRnd (https://shiny.cnsgenomics.com/mRnd/) online MR power calculator. This website relies on the research conducted by Brion MJ et al. [33], who utilized a noncentrality parameter-based method to evaluate MR statistical power.

### Sensitivity analysis

A variety of methods were employed to evaluate the sensitivity and stability of the results. First, Cochran's Q test was used to detect potential heterogeneity by the IVW and MR–Egger methods. Subsequently, the MR–Egger intercept test and Mendelian Randomization Pleiotropy RESidual Sum and Outlier (MR-PRESSO) global test were used to investigate the presence of horizontal pleiotropy. Additionally, the MR-PRESSO outlier and distortion test could also identify the outliers in the associations and correct for horizontal pleiotropy after excluding these outliers [34], and we also reperformed the MR analysis after excluding the outliers. Finally, we performed leave-one-out sensitivity test to assess the stability and validity of the causal effect estimates by excluding IVs one at a time.

All statistical analyses were executed using R software (version 4.2.1) with the R packages "Two sample MR", "forest plot" and "MRPRESSO". $P < 0.05$ indicated statistical significance of causal associations. This study did not need ethical approval, because it was a secondary analysis based on the original research, and each original study used in this analysis obtained informed consent from the participants.

## Results

### Selection of IVs

After selecting SNPs that were genome-wide independent ($r^2 < 0.001$, clumping window size = 10000 kb) and significant ($P < 5 \times 10^{-8}$), 76, 38 and 49 SNPs were selected as IVs for RA, cataract and glaucoma in the European population, respectively. Notably, the lowest F-statistic among these SNPs was 237.714, indicating minimal risk of weak instrumental bias. Detailed information on the F-statistic for each SNP can be found in S1 Table. Subsequently, the PhenoScanner V2 website was employed to identify and remove SNPs that were associated with outcome and confounders. Notably, no SNP was excluded during this step. Finally, we harmonized the datasets for exposure and outcome, and SNPs with palindromic intermediate allele frequencies were removed for further MR analysis. In the East Asian population, the selection process of IVs was consistent with the above steps. After rigorous programmatic screening, the number of selected IVs ranged from 8 to 56. A comprehensive list of all IVs used in this study is provided in S2 Table.

### Causal associations between RA with cataract and glaucoma in the European population

The MR analysis results employing various methods are depicted in Fig 2 and S1 Fig. The analysis results of IVW genetically indicated that RA was positively associated with cataract

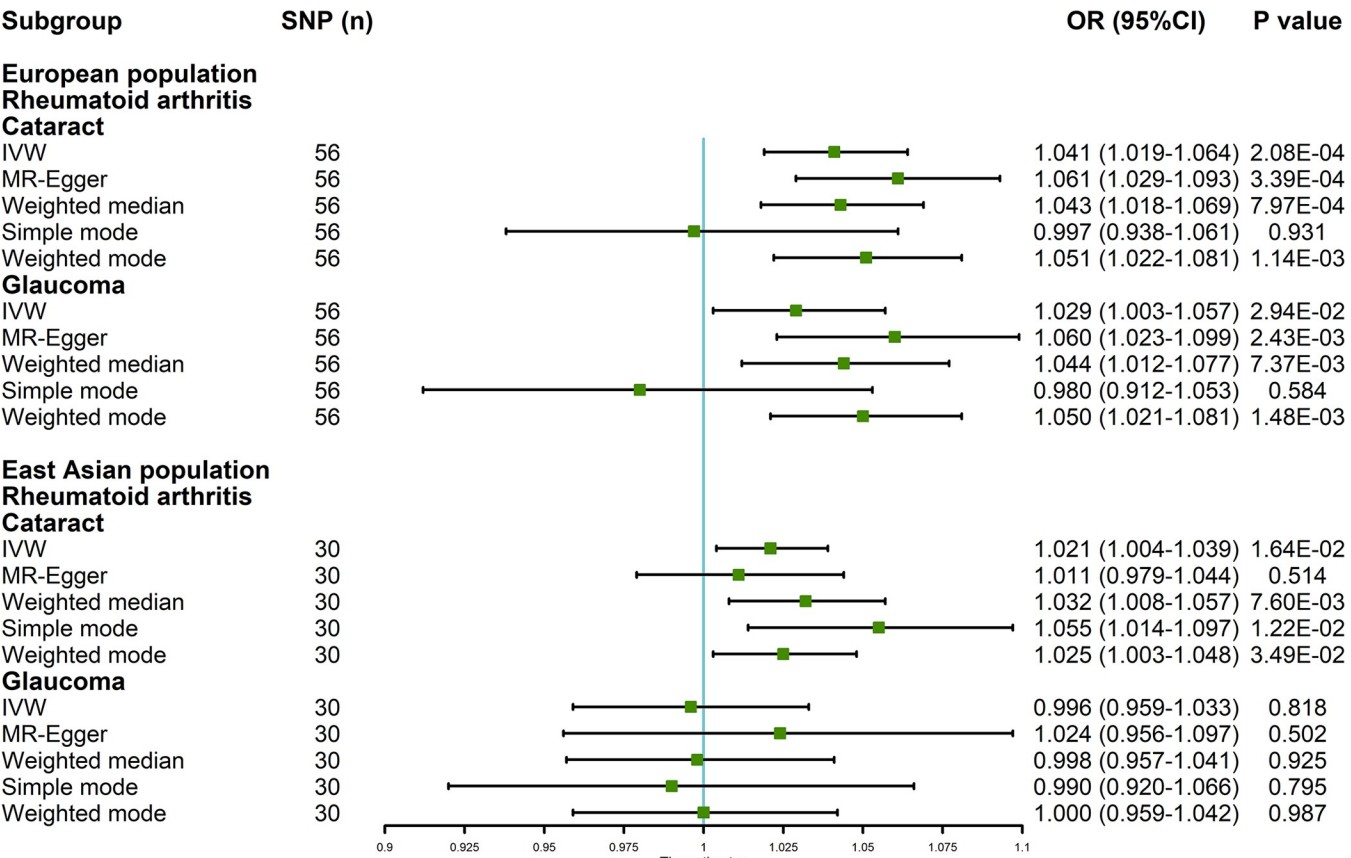

**Fig 2. MR analysis results of the causal effect of RA on cataract and glaucoma in European and East Asian ancestry.** RA: rheumatoid arthritis; OR: odds ratio; CI: confidence interval; IVW: inverse variance weighted.

(OR = 1.041, 95% confidence interval [CI] = 1.019, 1.064; P = 2.08×10$^{-4}$) and glaucoma (OR = 1.029, 95% CI = 1.003, 1.057; P = 2.94×10$^{-2}$) in the European population (Fig 2). In the reverse MR analyses, the IVW results confirmed that cataract and glaucoma had no causal effect on RA (both P>0.05) in S1 Fig. These findings were consistently supported by other analytical methods, and scatter plots visually reinforced these results (Fig 3). The statistical power of the IVW results can be found in S3 Table, with the lowest statistical power observed at 90% (RA on glaucoma).

Notably, Cochran's Q test revealed the presence of heterogeneity in all associations (all P<0.05). Then, the Egger intercept provided no evidence supporting horizontal pleiotropy in all analysis results (P>0.05), except for the association of RA on glaucoma (P = 0.029). The MR-PRESSO global test suggested the existence of horizontal pleiotropy in all associations (all P values of MR-PRESSO global test<0.05). Moreover, the MR-PRESSO outlier test identified a series of significant outliers in these associations (S4 Table), and we conducted additional MR analyses after removing these outliers. The IVW method showed that our analysis results were stable and reliable. The distribution of IVs was symmetric in funnel plots after excluding the outliers (S2 Fig), and both heterogeneity and horizontal pleiotropy were no longer present (P>0.05). The summarized sensitivity analysis results are presented in Table 2. Finally, the leave-one-out sensitivity tests indicated that most of the analysis results were stable and reliable (S3 Fig).

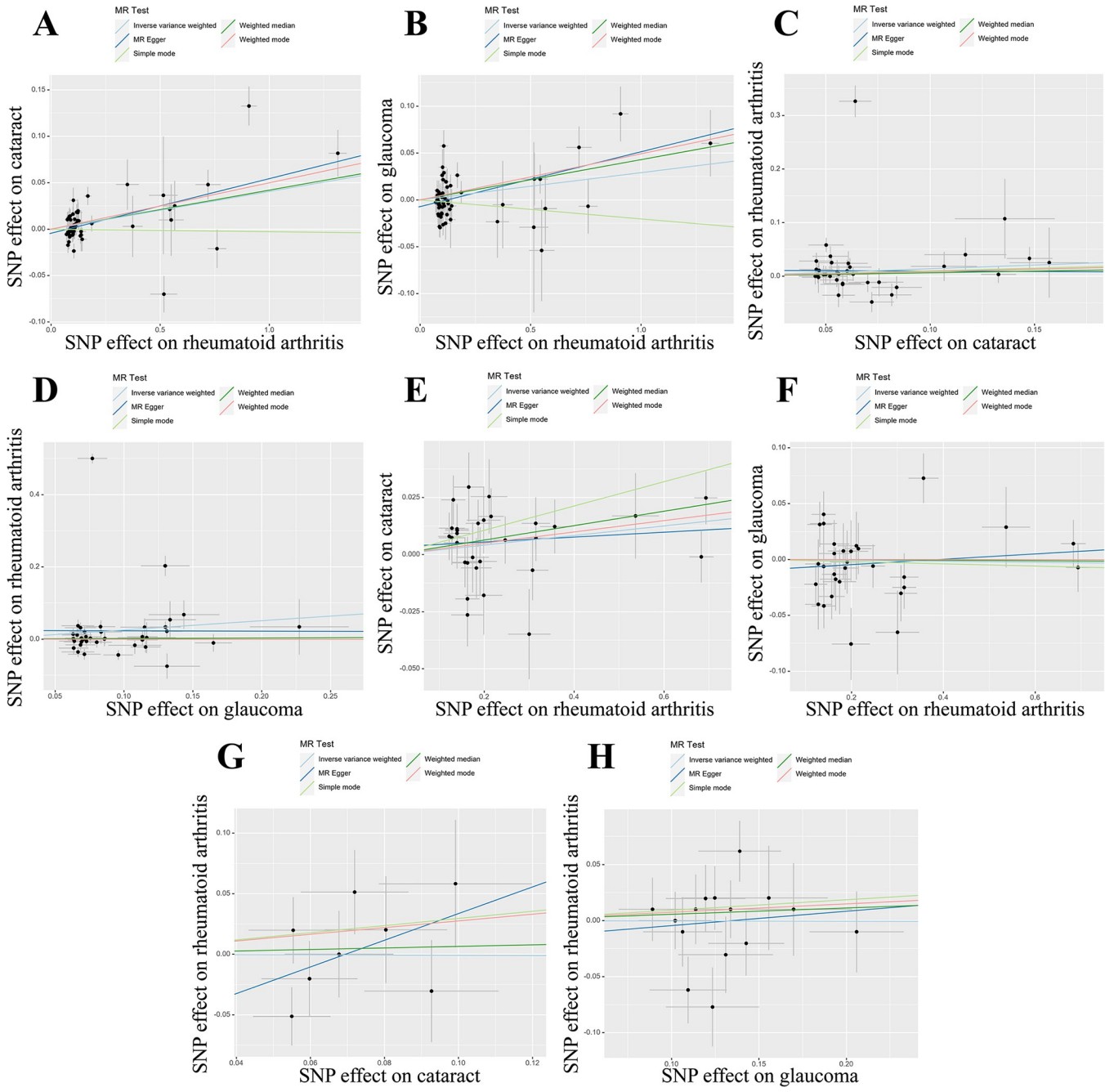

**Fig 3. Scatter plots for the causal associations of RA with cataract and glaucoma.** The ordinate in scatter plots is the effect of the instrumental variables on the outcome, and the abscissa is the effect of the instrumental variables on the exposure. The causal effect of each method is represented by the slope of the line. (A) Scatter plot of RA on cataract in the European population; (B) Scatter plot of RA on glaucoma in the European population; (C) Scatter plot of cataract on RA in the European population; (D) Scatter plot of glaucoma on RA in the European population; (E) Scatter plot of RA on cataract in the East Asian population; (F) Scatter plot of RA on glaucoma in the East Asian population; (G) Scatter plot of cataract on RA in the East Asian population; (H) Scatter plot of glaucoma on RA in the East Asian population. RA: rheumatoid arthritis.

**Table 2. The sensitivity analysis results in European and East Asian ancestry.**

| Population | Exposure | Outcome | Cochran's Q test | | Egger Intercept test | | MR-PRESSO | |
|---|---|---|---|---|---|---|---|---|
| | | | Q-IVW | P value | Egger-intercept | P value | P value of global test | P value of distortion test |
| European ancestry | RA | Cataract# | 67.961 | 0.068 | -0.003 | 0.208 | 0.059 | NA |
| | RA | Cataract* | 121.891 | $5.65*10^{-07}$ | -0.004 | 0.102 | <0.001 | 0.533 |
| | RA | Glaucoma# | 57.251 | 0.254 | -0.005 | 0.068 | 0.204 | NA |
| | RA | Glaucoma* | 91.247 | 0.002 | -0.007 | 0.029 | 0.001 | NA |
| | Cataract | RA# | 26.261 | 0.662 | 0.001 | 0.943 | 0.665 | NA |
| | Cataract | RA* | 167.709 | $5.88*10^{-20}$ | 0.011 | 0.591 | <0.001 | 0.075 |
| | Glaucoma | RA# | 40.370 | 0.177 | -0.001 | 0.964 | 0.209 | NA |
| | Glaucoma | RA* | 1429.142 | $1.75*10^{-275}$ | 0.024 | 0.711 | <0.001 | <0.001 |
| East Asian ancestry | RA | Cataract* | 34.647 | 0.216 | 0.003 | 0.462 | 0.175 | NA |
| | RA | Glaucoma# | 34.708 | 0.179 | -0.009 | 0.322 | 0.235 | NA |
| | RA | Glaucoma* | 46.543 | 0.021 | -0.009 | 0.346 | 0.030 | 0.593 |
| | Cataract | RA* | 9.614 | 0.211 | -0.077 | 0.297 | 0.362 | NA |
| | Glaucoma | RA* | 17.308 | 0.240 | -0.017 | 0.712 | 0.266 | NA |

RA: rheumatoid arthritis; IVW: inverse variance weighted; MR-PRESSO: mendelian randomization pleiotropy Residual Sum and Outlier. #: The sensitivity analysis after removing the outliers identified by the MR-PRESSO test. *: The sensitivity analysis before removing the outliers identified by the MR-PRESSO test.

## Causal associations between RA with cataract and glaucoma in the East Asian population

We also conducted MR analysis in the East Asian population, and the results by various methods are summarized in Fig 2 and S1 Fig. In the East Asian population, the IVW method genetically predicted that RA was associated with an elevated risk of cataract (OR = 1.021, 95% CI = 1.004, 1.039; P = $1.64×10^{-2}$), which is consistent with other methods. The statistical power of the IVW analysis result was 71% (S3 Table). However, the IVW result showed that RA had no significant effect on glaucoma (P = 0.818, Fig 2). Additionally, reverse MR analyses indicated that cataract and glaucoma were not positively associated with RA (both P>0.05, S1 Fig), consistent with other methods. Scatter plots provided a more intuitive representation of these results in Fig 3.

Notably, heterogeneity was not observed in these associations according to Cochran's Q test (all P>0.05), except for the association of RA on glaucoma (P = 0.021). The Egger intercept also provided no evidence of horizontal pleiotropy in all analysis results (all P>0.05). The MR-PRESSO global test found no evidence of horizontal pleiotropy in most associations, except the connection of RA on glaucoma (P of MR-PRESSO global test = 0.030). Moreover, the MR-PRESSO outlier test identified rs3819720 as an outlier in the association of RA on glaucoma. Following the exclusion of this outlier, the IVW analysis result remained stable, with no observed horizontal pleiotropy or heterogeneity (P>0.05). The distribution of IVs in funnel plots was symmetric, as shown in S2 Fig. The final results of sensitivity analysis are shown in Table 2. Finally, the leave-one-out tests demonstrated that excluding any single SNP did not substantially influence the causal associations (S3 Fig), which indicated that our results were stable and reliable.

## Discussion

This study systematically evaluated the genetic causal associations between RA with cataract and glaucoma in both European and East Asian populations. The GWAS summary data for cataract and glaucoma in European populations were derived from the 9th version of the

FinnGen GWAS summary statistics, while summary data for RA were obtained from the European GWAS catalog (http://www.ebi.ac.uk/). For the East Asian population, GWAS summary data for cataract, glaucoma and RA were all extracted from the IEU Open GWAS project (http://gwas.mrcieu.ac.uk/). All IVs underwent a rigorous selection process, resulting in the identification of 8 to 56 SNPs that were considered suitable for MR analysis. The IVW results indicated a positive genetic association between RA and an increased risk of cataract and glaucoma in the European population. Furthermore, RA exhibited a slightly increased risk of cataract in the East Asian population. However, reverse MR analyses suggested that cataract and glaucoma had no causal effect on RA. These findings provide valuable insights into early prevention strategies for cataract and glaucoma in individuals with RA and underscore the influence of RA-induced inflammation on ophthalmic diseases.

RA is a significant contributor to joint pain and dysfunction, particularly among the elderly, with its incidence on the rise each year [35]. It is characterized by inflammatory changes in the synovial membrane of joints and erosive arthritis [36]. Oxidative stress has gained increasing attention in recent decades as a key player in the development of RA [37]. Elevated levels of reactive oxygen species, a prominent biomarker of oxidative stress, have been observed in both the mitochondria and blood of RA patients [38, 39]. These reactive oxygen species can directly or indirectly cause damage to articular cartilage, leading to the degradation of proteoglycans and the inhibition of their synthesis [40]. Oxidative stress is also recognized as a driving factor in the pathogenesis of cataract, where an imbalance in the lens's redox state, driven by oxidative stress, contributes to cataract development [5, 41, 42]. Additionally, oxidative stress accelerates the loss of lens epithelial cells, which is a critical factor in cataract development [4, 43]. Oxidative stress plays a vital role in the development of both RA and cataract. Previous studies have reported a connection between RA and cataract. For example, Akintayo RO et al. observed cataract in 26% of RA patients in a study involving 50 individuals from the African population [44], and Black RJ et al. posited that RA could contribute to cataract development [45]. Consistent with these previous studies, our study confirmed a heightened risk of cataract in individuals with RA in European and East Asian populations, and these analysis results were credible and stable. We can summarize that RA can increase the risk of cataract on the basis of the above results. Interestingly, while a recent random forest analysis suggested that cataract might increase the risk of RA among individuals of European ancestry [14], our findings revealed that cataract had no causal effect on RA in both European and East Asian populations. This discrepancy could potentially be attributed to the influence of confounding factors such as aging and smoking. Therefore, it is essential to further validate this result in the future.

Additionally, local inflammation is a central element in the development of RA. Inflammatory factors and chemokines, including tumor necrosis factor (TNF), interleukin (IL) and matrix metalloproteinase (MMP), are upregulated in the synovial macrophages and dendritic cell subsets of RA patients [11, 46]. These inflammatory mediators lead to cartilage degradation, bone erosion, and the acceleration of RA development [47]. Similarly, inflammation plays a crucial role in the pathogenesis of glaucoma [48]. Inflammatory factors, such as TNF, IL-1β, IL-18 and MMP, can promote the death of RGCs, which is a hallmark of glaucoma development [48, 49]. Several previous studies have explored the associations between RA and glaucoma. For instance, Kim SH et al. conducted a propensity-matched cohort study involving the Korean population, revealing that RA could elevate the risk of open-angle glaucoma [15]. However, our MR analysis contradicted Kim SH et al.'s findings. We observed no causal effect of RA on glaucoma in the East Asian population. In contrast, we found a positive association between RA and glaucoma in the European population. Given these differing results, it is essential to emphasize the need for further validation of the association between RA and glaucoma. Interestingly, the findings from our MR analysis align with those of the previous

random forest analysis, which showed that glaucoma is not causally linked to an elevated risk of RA [14]. This consistency underscores that glaucoma does not actively promote the development of RA in individuals of European and East Asian ancestry.

A main advantage of this study design is that the causal associations are not disturbed by potential confounders and reverse causality. In addition, multiple sensitivity analyses were used to guarantee that the MR analysis results were stable in this study. However, there are also several limitations in the current MR study. Initially, it is worth noting that several insignificant analysis results in this study exhibit relatively low MR statistical power, and a cautious approach is warranted when interpreting these conclusions. Additionally, we performed the MR analysis using only European and East Asian ancestry. Therefore, it should be applied cautiously to other populations. In addition, due to the inherent characteristics of the GWAS data, detailed clinical information on participants, including specifics on steroid use in RA patients, was not available. Consequently, we were unable to conduct further subgroup analyses to evaluate the potential impact of associated drug use on the occurrence of glaucoma and cataract. Moreover, the pathogenesis of RA causing the development of cataract and glaucoma is still unclear and needs more research in the future.

## Conclusions

Our MR analyses revealed that RA was a risk factor for cataract and glaucoma. Moreover, cataract and glaucoma had no effect on RA in reverse MR analyses. We believe that oxidative stress and local inflammation are responsible for these causal associations. The results of this study can offer guidance on the early prevention of cataract and glaucoma in RA patients and provide some evidence for the influence of RA-induced inflammation on ophthalmic diseases.

## Supporting information

**S1 Fig. MR analysis results of the causal effect of cataract and glaucoma on RA in European and East Asian ancestry.** SNP (n): the number of single-nucleotide polymorphisms; RA: rheumatoid arthritis; OR: odds ratio; CI: confidence interval; IVW: inverse variance weighted.
(TIF)

**S2 Fig. Funnel plots for the causal associations of RA with cataract and glaucoma in European and East Asian ancestry.** The symmetry of the distribution of IVs can reflect the existence of horizontal pleiotropy to some extent. (A) Funnel plot of RA on cataract in the European population; (B) Funnel plot of RA on glaucoma in the European population; (C) Funnel plot of cataract on RA in the European population; (D) Funnel plot of glaucoma on RA in the European population; (E) Funnel plot of RA on cataract in the East Asian population; (F) Funnel plot of RA on glaucoma in the East Asian population; (G) Funnel plot of cataract on RA in the East Asian population; (H) Funnel plot of glaucoma on RA in the East Asian population. RA: rheumatoid arthritis; IVs: instrumental variables.
(TIF)

**S3 Fig. The leave-one-out sensitivity tests of the causal associations of RA with cataract and glaucoma in European and East Asian ancestry.** (A) Leave-one-out sensitivity test of RA on cataract in the European population; (B) Leave-one-out sensitivity test of RA on glaucoma in the European population; (C) Leave-one-out sensitivity test of cataract on RA in the European population; (D) Leave-one-out sensitivity test of glaucoma on RA in the European population; (E) Leave-one-out sensitivity test of RA on cataract in the East Asian population; (F) Leave-one-out sensitivity test of RA on glaucoma in the East Asian population; (G) Leave-one-out sensitivity test of cataract on RA in the East Asian population; (H) Leave-one-out

sensitivity test of glaucoma on RA in the East Asian population. RA: rheumatoid arthritis.
(TIF)

**S1 Table. The F statistics of SNPs in MR analyses.** SNP: single-nucleotide polymorphism;
Beta: estimate coefficient; P value: P value from GWAS; SE: standard error of coefficient estimate; R2: the variance of exposure explained by each IV.
(XLSX)

**S2 Table. Detailed information on instrumental variables used in MR analyses.** SNP: single-nucleotide polymorphism; Beta: Estimate coefficient; P value: P value from GWAS; SE:
standard error of coefficient estimate.
(XLSX)

**S3 Table. The statistical power of IVW results in this study.** IVW: inverse variance
weighted.
(XLSX)

**S4 Table. The SNPs eliminated by MR-PRESSO outlier test.** SNP: single-nucleotide polymorphism; MR-PRESSO: Mendelian Randomization Pleiotropy RESidual Sum and Outlier.
(XLSX)

## Acknowledgments

All genetic summary data were obtained from FinnGen, GWAS catalog and IEU Open GWAS
project. We thank all the participants and coordinators for the data used in this study.

## Author Contributions

**Conceptualization:** Menghao Teng, Jiachen Wang.

**Formal analysis:** Menghao Teng, Jiachen Wang.

**Investigation:** Ye Tian.

**Project administration:** Yingang Zhang.

**Software:** Xiaochen Su, Jiqing Wang.

**Writing – original draft:** Menghao Teng, Jiachen Wang.

**Writing – review & editing:** Menghao Teng, Jiachen Wang.

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
