## [Decision Letter · Decision Letter 0]

6 Dec 2023

PONE-D-23-32641Causal associations between rheumatoid arthritis and cataract and glaucoma in European and East Asian populations: a bidirectional two-sample mendelian randomization studyPLOS ONE

Dear Dr. Zhang,

Thank you for submitting your manuscript to PLOS ONE. After careful consideration, we feel that it has merit but does not fully meet PLOS ONE’s publication criteria as it currently stands. Therefore, we invite you to submit a revised version of the manuscript that addresses the points raised during the review process.

**ACADEMIC EDITOR: **Please attend to reviewers' comments.==============================

We look forward to receiving your revised manuscript.

Kind regards,

Nader Hussien Lotfy Bayoumi, M.D., FRCS (Glasgow)

Academic Editor

PLOS ONE

Journal Requirements:

"This work was supported by the Shaanxi Provincial Administration of Traditional Chinese Medicine (No. 2021-04-ZZ-003). Financial support had no impact on the outcomes of this study."

Reviewers' comments:

Reviewer's Responses to Questions

**Comments to the Author**

1. Is the manuscript technically sound, and do the data support the conclusions?

Reviewer #1: Yes

Reviewer #2: Yes

2. Has the statistical analysis been performed appropriately and rigorously? 

Reviewer #1: I Don't Know

Reviewer #2: Yes

3. Have the authors made all data underlying the findings in their manuscript fully available?

Reviewer #1: Yes

Reviewer #2: Yes

4. Is the manuscript presented in an intelligible fashion and written in standard English?

Reviewer #1: Yes

Reviewer #2: Yes

5. Review Comments to the Author

Reviewer #1: Title

I suggest removing the word “and” before cataract so as to be

‘Causal associations between rheumatoid arthritis, cataract and glaucoma in European and East Asian populations: a bidirectional two-sample mendelian randomization study’

Introduction

Page 4 line 47,48: “Currently, the global annual incidence rate of RA is 0.03%, 48 with a prevalence rate of 1%”. The reference used is 2016 which is not recent at all.

Materials and Methods

The number of cases and controls in the east Asian population is significantly lower than European ancestry. Also, the years of selection are in 2 different decades.

Results

The results of sensitivity tests written in text are different from table 2.

Examples:

Page 11 line 201-205:

“Notably, Cochran's Q test revealed the presence of heterogeneity in all associations 202 (all P<0.05).

The MR-PRESSO global test suggested the existence of horizontal pleiotropy in all associations (all P values of MR-PRESSO global test<0.05)”.

While in table 2, results of p-values are different.

Page 13 line 230-234:

“Notably, heterogeneity was not observed in these associations according to Cochran's 230 Q test (all P>0.05), except for the association of RA on glaucoma (P=0.021). The Egger 231 intercept also provided no evidence of horizontal pleiotropy in all analysis results (all 232 P>0.05). The MR-PRESSO global test found no evidence of horizontal pleiotropy in 233 most associations, except the connection of RA on glaucoma (P of MR-PRESSO global 234 test=0.030)”

Where are these p-values in table 2?

If results in table 2 are the finalized results and different from that of text plz clarify this

Discussion

I would like to thank the authors for these detailed and well-explained discussion. However, there are many confounding factors that could contribute to development of cataract and glaucoma in RA patients e.g., steroid intake in these patients which is known cause of cataract and glaucoma and daily doses can differ between patients. So, absence of assessment of drug history in these patients could have biased the results.

Page 15 line 282: “However, it is essential to underscore that further validation of this result is imperative”. This sentence needs re-phrasing for more clarification.

Page 16 line 289, 290: “Inflammatory factors such as TNF, IL and MMP can promote the death of RGCs, which is a hallmark of glaucoma development [40, 41]”

Which ILs are involved in development of glaucoma? In comparison to those responsible for RA.

The full name of RGCS should be mentioned.

Reviewer #2: I am pleased to review the manuscript. Here are some comments to the authors:

1. Abstract: It is recommended to show the data sources and sample sizes of exposures and outcomes in this section. In addition, the authors are also suggested to add the number of SNPs in this part.

2. Introduction, lines 60-63: Need references for the content.

3. Introduction: In this part, the authors are recommended to show some previous Mendelian randomization (MR) studies related to rheumatoid arthritis. For example, PMID: 37039764, 34015765, 35303571, and 36514134.

4. Materials and methods: A STROBE-MR checklist (PMID: 34702754) will be useful for readers to know the components of the current MR study.

5. IV selection: To ensure the SNPs were not related to potential covariates, sometimes we can select SNPs that were exclusively associated with the exposures. Such a method can be added a sensitivity analysis for the present study.

6. Statistical analysis: The authors should clearly state what types of IVW they used (fixed-effects or random-effects).

7. Discussion, lines 257-259: Need references for the content.

6. PLOS authors have the option to publish the peer review history of their article (what does this mean?). If published, this will include your full peer review and any attached files.

Reviewer #1: **Yes: **Sally S Mohamed

Reviewer #2: No

---

## [Author Response · Author response to Decision Letter 0]

8 Jan 2024

Response to Reviewers

 Submission Number: PONE-D-23-32641

Title of Paper: Causal associations between rheumatoid arthritis and cataract and glaucoma in European and East Asian populations: a bidirectional two-sample mendelian randomization study

Dear Editors,

 Thank you for giving us the opportunity to submit a revised draft of our manuscript. We appreciate the time and effort that you and the reviewers have dedicated to providing valuable feedback. We are grateful to the reviewers for their insightful comments. We addressed each of the points raised by the reviewers and modified the manuscript accordingly. All changes are highlighted in Yellow.

Response to Editor

 According to your suggestions, we have made the following changes in the revised manuscript: 1: We revised our manuscript to meet PLOS ONE’s style requirements. 2: We state what role the funders took in the study, “The funders had no role in study design, data collection and analysis, decision to publish, or preparation of the manuscript”. 3: We provide the correct grant numbers for the awards we received for our study in the ‘Funding Information’ section. 

Reviewer 1 Comments

1. Title

I suggest removing the word “and” before cataract so as to be

‘Causal associations between rheumatoid arthritis, cataract and glaucoma in European and East Asian populations: a bidirectional two-sample mendelian randomization study’

Response: Thank you for your thoughtful and constructive suggestion regarding the title.

 We appreciate your attention to detail. Following your guidance, we have revised the title as recommended, and it now reads: “Causal associations between rheumatoid arthritis, cataract and glaucoma in European and East Asian populations: a bidirectional two-sample mendelian randomization study”

2. Introduction

Page 4 line 47,48: “Currently, the global annual incidence rate of RA is 0.03%, 48 with a prevalence rate of 1%”. The reference used is 2016 which is not recent at all.

Response: Thank you for your constructive suggestions.

 I appreciate your attention to detail and the opportunity to clarify this aspect of our presentation. We have made a correction in the revised manuscript.

 Location: Introduction section, Line 53-54: “As of 2016, the global annual incidence rate of RA is 0.03%, with a prevalence rate of 1% [12].”

Reference: 

12. Smolen JS, Aletaha D, McInnes IB. Rheumatoid arthritis. Lancet. 2016;388(10055):2023-38.

3. Materials and Methods

The number of cases and controls in the east Asian population is significantly lower than European ancestry. Also, the years of selection are in 2 different decades

Response: Thank you for your constructive comments.

 We appreciate your insightful observation regarding the limitations in the number of cases and controls within the East Asian population, as well as the variations in the years of selection spanning two different decades. The scarcity of open databases for East Asian populations presented a challenge, leading us to utilize the GWAS summary data available in the IEU OPEN GWAS project. It is essential to acknowledge the constraints inherent in the East Asian databases, characterized by older data, limited sample size, and non-synchronicity in data collection across the decades. Moving forward, we aspire to conduct further analyses with newer and more extensive databases to strengthen the robustness of our study.

4. Results

The results of sensitivity tests written in text are different from table 2.

Examples:

Page 11 line 201-205:

“Notably, Cochran's Q test revealed the presence of heterogeneity in all associations 202 (all P<0.05).

The MR-PRESSO global test suggested the existence of horizontal pleiotropy in all associations (all P values of MR-PRESSO global test<0.05)”.

While in table 2, results of p-values are different.

Page 13 line 230-234:

“Notably, heterogeneity was not observed in these associations according to Cochran's 230 Q test (all P>0.05), except for the association of RA on glaucoma (P=0.021). The Egger 231 intercept also provided no evidence of horizontal pleiotropy in all analysis results (all 232 P>0.05). The MR-PRESSO global test found no evidence of horizontal pleiotropy in 233 most associations, except the connection of RA on glaucoma (P of MR-PRESSO global 234 test=0.030)”

Where are these p-values in table 2?

If results in table 2 are the finalized results and different from that of text plz clarify this

Response: Thank you very much for your valuable comments and careful review of our manuscript.

 I sincerely apologize for any confusion caused by the discrepancy between the results presented in the manuscript and Table 2 regarding the sensitivity tests. To address this concern, we have made the necessary clarification in the revised manuscript. The results presented in Table 2 now include both the results after removing outliers and before such removal, providing a comprehensive overview of the sensitivity analysis. 

 Location: Results section, Line 228-232: “

Table 2: The sensitivity analysis results in European and East Asian ancestry.

Population Exposure Outcome Cochran's Q test Egger Intercept test MR-PRESSO

 Q-IVW P value Egger-intercept P value P value of global test P value of distortion test

European ancestry RA Cataract# 67.961 0.068 -0.003 0.208 0.059 NA

 RA Cataract* 121.891 5.65*10-07 -0.004 0.102 <0.001 0.533

 RA Glaucoma# 57.251 0.254 -0.005 0.068 0.204 NA

 RA Glaucoma* 91.247 0.002 -0.007 0.029 0.001 NA

 Cataract RA# 26.261 0.662 0.001 0.943 0.665 NA

 Cataract RA* 167.709 5.88*10-20 0.011 0.591 <0.001 0.075

 Glaucoma RA# 40.370 0.177 -0.001 0.964 0.209 NA

 Glaucoma RA* 1429.142 1.75*10-275 0.024 0.711 <0.001 <0.001

East Asian ancestry RA Cataract* 34.647 0.216 0.003 0.462 0.175 NA

 RA Glaucoma# 34.708 0.179 -0.009 0.322 0.235 NA

 RA Glaucoma* 46.543 0.021 -0.009 0.346 0.030 0.593

 Cataract RA* 9.614 0.211 -0.077 0.297 0.362 NA

 Glaucoma RA* 17.308 0.240 -0.017 0.712 0.266 NA

RA: rheumatoid arthritis; IVW: inverse variance weighted; MR-PRESSO: mendelian randomization pleiotropy Residual Sum and Outlier. #: The sensitivity analysis after removing the outliers identified by the MR-PRESSO test. *: The sensitivity analysis before removing the outliers identified by the MR-PRESSO test.”

5. Discussion

I would like to thank the authors for these detailed and well-explained discussion. However, there are many confounding factors that could contribute to development of cataract and glaucoma in RA patients e.g., steroid intake in these patients which is known cause of cataract and glaucoma and daily doses can differ between patients. So, absence of assessment of drug history in these patients could have biased the results.

Response: Thank you for raising a crucial concern regarding potential confounding factors, particularly steroid intake in RA patients and their impact on the development of cataract and glaucoma.

 We genuinely appreciate your insightful observation. Unfortunately, due to the limitations of available GWAS data, detailed clinical information, including medication history, was not accessible. Consequently, we were unable to conduct subgroup analyses to specifically address the influence of steroid medication on the occurrence of cataract and glaucoma in RA patients.

 In our IVs selection process, we diligently excluded known confounding factors such as aging, smoking, and steroid use to the best extent possible. However, we acknowledge the limitation imposed by the lack of detailed medication history.

 Location: Discussion section, Line 326-331: “In addition, due to the inherent characteristics of the GWAS data, detailed clinical information on participants, including specifics on steroid use in RA patients, was not available. Consequently, we were unable to conduct further subgroup analyses to evaluate the potential impact of associated drug use on the occurrence of glaucoma and cataract.”

Page 15 line 282: “However, it is essential to underscore that further validation of this result is imperative”. This sentence needs re-phrasing for more clarification.

Response: Thank you for your insightful feedback.

 I appreciate your observation regarding the clarity of the sentence in question. In response to your suggestion, we have revised the sentence in the discussion section for improved clarity.

 Location: Discussion section, Line 297-298: “Therefore, it is essential to further validate this result in the future.”

Page 16 line 289, 290: “Inflammatory factors such as TNF, IL and MMP can promote the death of RGCs, which is a hallmark of glaucoma development [40, 41]”

Which ILs are involved in development of glaucoma? In comparison to those responsible for RA.

The full name of RGCS should be mentioned.

Response: Thank you for providing valuable feedback.

 I sincerely appreciate your insightful comments regarding our manuscript. I apologize for the oversight in not explicitly mentioning which interleukins involved in the development of glaucoma. In line with your suggestion, we have made the necessary additions to the revised manuscript to address this omission.

 Location: Results section, Line 305-306: “Inflammatory factors, such as TNF, IL-1β, IL-18 and MMP, can promote the death of RGCs, which is a hallmark of glaucoma development [48, 49].”

Reference: 

48. Vohra R, Tsai JC, Kolko M. The role of inflammation in the pathogenesis of glaucoma. Surv Ophthalmol. 2013;58(4):311-20.

49. Fontaine V, Mohand-Said S, Hanoteau N, Fuchs C, Pfizenmaier K, Eisel U. Neurodegenerative and neuroprotective effects of tumor Necrosis factor (TNF) in retinal ischemia: opposite roles of TNF receptor 1 and TNF receptor 2. J Neurosci. 2002;22(7):Rc216.

 Moreover, I would like to assure you that the full name of RGCs, namely retinal ganglion cells, has already been provided in the manuscript, specifically in the Introduction section, Line 41.

Reviewer 2 Comments

1. Abstract: It is recommended to show the data sources and sample sizes of exposures and outcomes in this section. In addition, the authors are also suggested to add the number of SNPs in this part.

Response: Thank you sincerely for your invaluable comments.

 I greatly appreciate your guidance. In response to your suggestions, we have incorporated information on data sources and sample sizes of exposures and outcomes, along with the number of SNPs, into the Abstract section of the revised manuscript.

 Location: Abstract section, Line 9-15: “In the European population, genome-wide association study (GWAS) summary statistics for cataract (372,386 individuals) and glaucoma (377,277 individuals) were obtained from the FinnGen consortium (R9), while RA summary data were derived from a meta-analysis of GWAS encompassing 97173 samples. In the East Asian population, summary data for cataract (212453 individuals), glaucoma (212453 individuals), and RA (22515 individuals) were sourced from the IEU Open GWAS project.” Line 22-23: “Following stringent screening, the number of selected instrumental variables ranged from 8 to 56.”

2. Introduction, lines 60-63: Need references for the content.

Response: Thank you for the helpful comments.

 As your guidance, we add the references in this part. After being revised, the manuscript has become rigorous.

 Location: Introduction section, Line 66-69: “In MR analysis, exposure is regarded as an intermediate phenotype, and single-nucleotide polymorphisms (SNPs) are selected as instrumental variables (IVs) to examine the causal associations between the exposure phenotype and disease outcome [17].”

Reference: 

17. Bowden J, Holmes MV. Meta-analysis and Mendelian randomization: A review. Res Synth Methods. 2019;10(4):486-96.

3. Introduction: In this part, the authors are recommended to show some previous Mendelian randomization (MR) studies related to rheumatoid arthritis. For example, PMID: 37039764, 34015765, 35303571, and 36514134.

Response: Thank you sincerely for your invaluable comments.

 I truly appreciate your guidance. In accordance with your suggestion, we have incorporated relevant MR studies related to rheumatoid arthritis into the Introduction section. Please see the revised manuscript.

 Location: Introduction section, Line 73-76: “Widely adopted, MR has been instrumental in exploring causal associations between RA and diverse diseases, spanning cancer [21], inflammatory bowel disease [22], and osteoporosis [23, 24].”

Reference: 

21. Yuan S, Chen J, Ruan X, Vithayathil M, Kar S, Li X, et al. Rheumatoid arthritis and risk of site-specific cancers: Mendelian randomization study in European and East Asian populations. Arthritis Res Ther. 2022;24(1):270.

22. Meisinger C, Freuer D. Rheumatoid arthritis and inflammatory bowel disease: A bidirectional two-sample Mendelian randomization study. Semin Arthritis Rheum. 2022;55:151992.

23. Deng Y, Wong MCS. Association Between Rheumatoid Arthritis and Osteoporosis in Japanese Populations: A Mendelian Randomization Study. Arthritis Rheumatol. 2023;75(8):1334-43.

24. Liu YQ, Liu Y, Chen ZY, Li H, Xiao T. Rheumatoid arthritis and osteoporosis: a bi-directional Mendelian randomization study. Aging (Albany NY). 2021;13(10):14109-30.

4. Materials and methods: A STROBE-MR checklist (PMID: 34702754) will be useful for readers to know the components of the current MR study.

Response: Thank you for providing constructive suggestions.

 I appreciate your guidance, and in response to your recommendation, we have included the STROBE-MR checklist in the Materials and Methods section of the manuscript. This addition aims to provide readers with a comprehensive understanding of the components of the current MR study.

 Location: Materials and methods section, Line 90-91: “This MR analysis meticulously adhered to the recommendations by the STROBE-MR [25].”

Reference: 

25. Skrivankova VW, Richmond RC, Woolf BAR, Davies NM, Swanson SA, VanderWeele TJ, et al. Strengthening the reporting of observational studies in epidemiology using mendelian randomisation (STROBE-MR): explanation and elaboration. Bmj. 2021;375:n2233.

5. IV selection: To ensure the SNPs were not related to potential covariates, sometimes we can select SNPs that were exclusively associated with the exposures. Such a method can be added a sensitivity analysis for the present study.

Response: Thank you for bringing up this important consideration.

 We appreciate your attention to the IVs selection process in MR analysis. To ensure the robustness of our approach, we employed a meticulous method. Initially, we selected SNPs that demonstrated significance (P<5×10-8 or P<5×10-6) and independence (r2<0.001, with a window size of 10000 kb) in relation to the exposure. Subsequently, we systematically excluded weak IVs with F-statistics exceeding 10. To further enhance the reliability of our IVs, we utilized the PhenoScanner V2 website (www.phenoscanner.medschl.cam.ac.uk) to meticulously exclude SNPs associated with outcome and potential confounders. As a result, all the final SNPs included in our analysis are robustly associated with the exposure, while demonstrating no significant associations with outcome variables and confounding factors.

6. Statistical analysis: The authors should clearly state what types of IVW they used (fixed-effects or random-effects).

Response: Thank you for your valuable comments.

 I appreciate your observation, and I apologize for the oversight in not explicitly specifying the type of IVW method utilized in our analysis. In this MR study, we employed the IVW (random-effects) method as the primary analysis. Additionally, to ensure a comprehensive assessment of causal associations between RA, cataract and glaucoma in European and East Asian populations, we complemented the primary analysis with MR‒Egger regression, weighted median, weighted mode, and simple mode methods.

 Location: Abstract section, Line 15-17: “Inverse-variance weighted (IVW, random-effects) method served as the primary analysis, complemented by MR‒Egger regression, weighted median, weighted mode and simple mode methods.” Materials and methods section, Line 144-147: “There were five complementary methods employed in this study to estimate the causal associations between RA and cataract and glaucoma, including inverse-variance weighted (IVW, random-effects), weighted median, MR‒Egger regression, simple mode and weighted mode methods.”

7. Discussion, lines 257-259: Need references for the content.

Response: Thank you for the helpful comments.

 As your guidance, we add the references in this section. Please see the revised manuscript.

 Location: Discussion section, Line 273-275: “RA is a significant contributor to joint pain and dysfunction, particularly among the elderly, with its incidence on the rise each year [35]. It is characterized by inflammatory changes in the synovial membrane of joints and erosive arthritis [36].”

Reference: 

35. Finckh A, Gilbert B, Hodkinson B, Bae SC, Thomas R, Deane KD, et al. Global epidemiology of rheumatoid arthritis. Nat Rev Rheumatol. 2022;18(10):591-602.

36. Zhou S, Lu H, Xiong M. Identifying Immune Cell Infiltration and Effective Diagnostic Biomarkers in Rheumatoid Arthritis by Bioinformatics Analysis. Front Immunol. 2021;12:726747.

Best wishes!

Yingang Zhang Ph.D., M.D.

Department of Orthopedics

The First Affiliated Hospital of Xi’an Jiaotong University

E-Mail: zyingang@mail.xjtu.edu.cn

---

## [Decision Letter · Decision Letter 1]

8 Feb 2024

Causal associations between rheumatoid arthritis, cataract and glaucoma in European and East Asian populations: a bidirectional two-sample mendelian randomization study

PONE-D-23-32641R1

Dear Dr. Zhang,

We’re pleased to inform you that your manuscript has been judged scientifically suitable for publication and will be formally accepted for publication once it meets all outstanding technical requirements.

Kind regards,

Nader Hussien Lotfy Bayoumi, M.D., FRCS (Glasgow)

Academic Editor

PLOS ONE

Additional Editor Comments (optional):

Thank you for addressing all reviewers' comments and concerns.

Reviewers' comments:

Reviewer's Responses to Questions

**Comments to the Author**

1. If the authors have adequately addressed your comments raised in a previous round of review and you feel that this manuscript is now acceptable for publication, you may indicate that here to bypass the “Comments to the Author” section, enter your conflict of interest statement in the “Confidential to Editor” section, and submit your "Accept" recommendation.

Reviewer #1: All comments have been addressed

Reviewer #2: All comments have been addressed

2. Is the manuscript technically sound, and do the data support the conclusions?

Reviewer #1: Yes

Reviewer #2: (No Response)

3. Has the statistical analysis been performed appropriately and rigorously? 

Reviewer #1: Yes

Reviewer #2: (No Response)

4. Have the authors made all data underlying the findings in their manuscript fully available?

Reviewer #1: Yes

Reviewer #2: (No Response)

5. Is the manuscript presented in an intelligible fashion and written in standard English?

Reviewer #1: Yes

Reviewer #2: (No Response)

6. Review Comments to the Author

Reviewer #1: I would like to thank the authors for their vigorous efforts and addressing all comments and presenting such comprehensive work

Reviewer #2: The authors’ responses have answered all earlier comments. I don't have any further comments on the current version.

7. PLOS authors have the option to publish the peer review history of their article (what does this mean?). If published, this will include your full peer review and any attached files.

Reviewer #1: **Yes: **Sally S Mohamed

Reviewer #2: No

---

## [Editor Report · Acceptance letter]

23 Feb 2024

PONE-D-23-32641R1 

PLOS ONE

Dear Dr. Zhang, 

I'm pleased to inform you that your manuscript has been deemed suitable for publication in PLOS ONE. Congratulations! Your manuscript is now being handed over to our production team.

Kind regards, 

on behalf of

Professor Nader Hussien Lotfy Bayoumi 

Academic Editor

PLOS ONE